# Behavioral Interventions in Long-Term Care Facilities during the COVID-19 Pandemic: A Case Study

**DOI:** 10.3390/geriatrics7010001

**Published:** 2021-12-21

**Authors:** Carlos Dosil-Díaz, David Facal, Romina Mouriz-Corbelle

**Affiliations:** 1Gerontological Therapeutic Complex “A Veiga”, Serge Lucense, Pobra de San Xiao, 27360 Lancara, Spain; romina.mouriz@usc.es; 2Department of Developmental Psychology, University of Santiago de Compostela, 15782 Santiago de Compostela, Spain; david.facal@usc.es

**Keywords:** mixed dementia, behavioral and psychological symptoms in dementia, challenging behaviors, lockdown

## Abstract

During the COVID-19 pandemic, long-term care (LTC) centers have adopted a series of measures that have affected the physical and cognitive health of patients. The routines of the patients, as well as the interventions of professionals, have been altered. In the case presented here, our aim was to explain the effect that the strong confinement due to the spread of the first COVID-19 wave in Spain had on a 75-year-old resident in an LTC center, with cognitive and behavioral symptomatology compatible with a diagnosis of mixed dementia, as well as the measures that the center adopted to manage the lockdown situation in the best possible way, including personalized attention protocols and a video call program. Different nosological hypotheses are also raised using a semiological analysis, including the analysis of the initial and continuation diagnostic protocols, as well as the therapeutic options.

## 1. Introduction

In March 2020, long-term care (LTC) centers were dramatically affected in their operations by the COVID-19 pandemic. The high rate of frailty of the old residents caused a disproportionate number of deaths during the first wave of the pandemic, reaching 20% in centers that were affected by COVID-19. The centers implemented a series of measures to deal with COVID-19, including the sectorization of spaces, use of PPE, controls of daily constants, reorganization of functions in the staff, and the restriction or elimination of visits from relatives to residents, among others [1]. Different studies show that the pandemic has had a very negative impact on institutionalized older people and, more specifically, on those who suffer cognitive impairment with behavioral alterations, due, to a large extent, to the fact that the restrictions on social relationships increase the problems of loneliness and isolation, which existed before the pandemic [2].

In LTC centers, 60–80% of patients have some degree of cognitive impairment, 20–30% show severe phase dementia, and 65% have some behavioral disorder [3]. In this way, patients suffering from dementia, in addition to presenting an important alteration in cognitive functions, due to the progressive deterioration of some brain functions, show the Behavioral and Psychological Symptoms of Dementia (BPSD). It has been described that between 60% and 90% of patients with dementia will present BPSD throughout the course of the disease [4]. This percentage varies depending on the stage and the previous personality of the patient. Behavioral symptoms refer to physical aggression, yelling, restlessness, wandering, culturally inappropriate behaviors, sexual disinhibition, harassment, inappropriate language, and persistent following of another. Psychological symptoms refer to anxiety problems, depressed mood, hallucinations and delusions, etc. [5]. One of the models used to address the management of behavior disorders is that of Cohen-Mansfield [6]. In this model, the needs of the person with dementia and professionals caring for them are considered, as well as clinical and environmental characteristics. The intervention, therefore, arises from a paradigm of personalized care, where the person is the center of the intervention, and the professional must adapt to the patient’s needs.

In this sense, it is important to highlight that LTC center residents often present disruptive symptoms, which increases the stress of the caregivers and the suffering of the person with dementia, sometimes reaching the need to use physical restraints and/or drugs to alleviate these symptoms [7]. For all these reasons, it is essential to carry out an interdisciplinary intervention focused on older residents. Although the tendency is to increase this type of tailorized interventions in LTC centers, during the COVID-19 pandemic, this has been hampered by the measures taken to contain the spread of the virus [8]. The present manuscript is a case study that addresses the intervention of a patient admitted to a residential LTC center during the first wave of the COVID-19 pandemic.

## 2. Materials and Methods

### 2.1. Case

#### 2.1.1. Personal History

The resident is 75 years old, a woman, and a widow, and she has three children. She was born in the rural area of Sarria, Lugo (Galicia, Spain). She entered the LTC center in January 2020. Until admission, she lived alone. She was a very active woman and ran a store for decades and she had a basic education.

#### 2.1.2. Medical and Psychological History

The resident presented dementia with a mixed profile and behavioral alterations, hypertensive heart disease, permanent atrial fibrillation, dyslipidemia, and osteoporosis. The information provided came from the reports of the relatives, as well as the social reports of the town hall and the clinical reports of the hospital.

In June 2018, a cranial CT scan was performed without intravenous contrast, showing no evidence of space-occupying lesions in an intraparenchymal or extra-axial location. There was supratentorial predominance of white matter, in periventricular and subcritical locations, which could correspond to leukopathy due to small vessel involvement. A revision was scheduled in September 2018 with analytical and EEG.

In August 2018, she was seen again by the specialist doctor, presenting the following symptoms: aggressiveness with the caregiver, shadowing, she had stopped cooking at home, and she was unable to perform simple memory tasks. The following treatment was prescribed: Bisprorol 2.5 mg, Digoxin 0.25, Atorvastatin 40 mg 0-0-1, Seguril 40 mg 1-0-0, Eliquis 5 mg 1-0-1, Deltius 25 mg, Rivastigmine 4.6 mg Doxium 2-0-0, and Irbesartan 300 mg 1-0-0.

In the psychological evaluation, she presented 6 errors in the Pfeiffer test, a MEC (a Spanish version of the Mini-Mental State Examination with a total score of 35) score of 21/35, and a Global Deterioration Scale (GDS) score of 4, scores that reflect moderate cognitive impairment.

In December 2018, the patient went to the hospital emergency department because she had had several syncope and falls at her home, with loss of consciousness.

During the year 2019, she stayed at her home supervised by her children, and in January 2020, she entered the A Veiga Xerontolóxico Therapeutic Complex, three months before the COVID-19 pandemic was declared in Spain.

#### 2.1.3. Status When Entering the LTC Center

Upon arrival, she was evaluated by the LTC center doctor in coordination with the referral geriatrician at the Lucus Augusti hospital. They observed that she presented flight delirium, anosognosia due to cognitive impairment, and independence in BADL without execution errors. Behavioral alterations stood out, with poor management with current treatment. An ECG was performed to rule out repolarization alterations in the face of blockages.

An adjustment of the medication was made taking into account the rejection that the patient presents to oral medicines. Seven drugs were maintained by eliminating Doxium and Irbesartan and adding two antipsychotics, Quetiapine, and a class of medication called NMDA receptor antagonists, Memantine.

In order for the psychological aspects to be evaluated, different cognitive screening tests were applied at the time of admission, including the MEC (a Spanish version of the Mini-Mental State Examination with a total score of 35), Global Deterioration Scale (GDS), and Clinical Dementia Rating (CDR) (Table 1), indicating that the resident is a person with severe cognitive impairment that includes impairment of memory, severe short-term memory circuit problems, severe temporal disorientation, and very basic spatial location, with impaired judgment, impaired problem solving, and impaired attention to personal care.

The patient preserved communicative and relational intentionality, compensating for certain cognitive deficits. She also preserved social verbal interaction. There were reports of unfinished sentences, occasional confusion of pronunciation patterns, and difficulty naming tasks. The content of her speech included fables and misinterpretation.

The Geriatric Assessment Scale (also called Yesavage scale) was used to assess mood, obtaining an inconclusive result (Table 1) given the unreliability of the responses given by the resident.

Regarding the physiotherapeutic evaluation, the resident maintained an independent and stable gait, without the need for support products. She presented good joint balance, without stiffness and maintaining functionality. She did not present alterations in muscle tone and had a perfect general muscular balance (Daniels 5/5). In terms of balance and gait, she had a Tinetti score of 16 in balance and 12 in running, which is a low-risk assessment. Regarding the risk of falls, in the up and go test, she presented a normal risk.

To assess the functionality of the patient, we applied the Barthel Index at the time of admission. Its result indicates a moderate dependence for BADL (Table 1), such as food, transfers, personal hygiene, use of the bathroom, personal hygiene, shower, mobility, going up or down stairs, and dressing and undressing.

Taking into account the above, the different departments of the center proposed the following objectives: maintain her level of independence in the performance of the ABVD; promote her communication and relationship with other residents; encourage her participation in activities; promote her adherence to routines and guidelines; maintain her motor, cognitive, and sensory performance skills that promote functional independence; promote cognitive stimulation activities; improve her temporal and spatial orientation; etc.

In order for the set objectives to be achieved, the patient was placed in a single room on the floor intended for patients with moderate dependence (called the Red Floor) whose routines were developed on the main floor of the center, giving them the possibility of accessing different activities.

### 2.2. Intervention during the Lockdown

In March 2020, with the arrival of COVID-19 in Spain and the state of alarm decreed by the government of Spain, the center was divided into three large areas, which prevented normal walking through it and interaction among residents, generating an increase in their behavioral disorders, such as agitation, disinhibition, repeated calls and questions, difficulty in cleaning and manifestation of a difficult temperament, mood alterations, demanding behaviors, and apathetic behaviors with a tendency for immobility [9].

Measures derived from the COVID-19 pandemic included the confinement of residents in their bedrooms or in restricted places close to their rooms, without contact with other residents and restricted contact with their professional caregivers. Accordingly, it was necessary to rethink activities in order to achieve the achieve the therapeutical aims. Below, we expose the activities in detail:-Attention was placed on the patient’s routines, with continuous follow-up and greater support for her BADL.-Due to the restrictions in the visitation regime (group activities and visits were suspended, according to the Decree of 17 March 2020, of the Xunta de Galicia), and since she was a person who socialized to a high degree with the residents and who had a high frequency of family visits, she was included in the center’s video call program, wherein the resident could interact with her family, thus stimulating social interaction (see Figure 1).-Personal care was also promoted, referring to hygiene, food, and clothing, as well as activities related to anti-COVID-19 measures (wearing a mask, hand disinfection). Pictograms and posters were used to facilitate these activities, wherein they were identified in a very simple way, for example, the room number with a photo of her face, posters on how to put on the mask and how to disinfect the hands, etc.

It should be noted that the patient, despite being in a sectorized area and of relative control, had serious difficulties staying in her area, mainly motivated by the cognitive impairment that she presented, with highly frequent ambulation and continuous attempts to establish contact with residents in her area and other areas.

In the last months of 2020 and early 2021, the patient experienced a worsening in BPSD (aggressiveness at bath time and personal hygiene, attempts to attack other residents, etc.). In order to guarantee adequate care for her, in March 2021, she was transferred to the psychogeriatric unit of the LTC center, which offers more specialized care that guarantees greater control, security, and supervision both for the patient herself and for the people with whom she had direct contact. In this unit, the monitoring by the medical, psychological, therapeutic, and care personnel is more intense and exhaustive. This unit has large areas, with wide corridors that favor walking. It also has a warm and homely design, with colors in light tones that help to maintain the tranquility of the residents, as well as relaxing background music. In the same way, the unit presents natural lighting, which provides a more pleasant stay. Despite being a unit where residents have high deterioration, it is about maintaining the functions and enhancing the autonomy of the residents.

Regarding the training of the personnel of the psychogeriatric unit, it should be noted that the staff have specific training that allows them to offer adequate care, emphasizing direct intervention, active roles, and feedback, despite having to work during the pandemic with a mask and protective screens. In this sense, the workers addressed the patient in a respectful way, understanding her pathology and enhancing her autonomy. The care staff of the unit were permanent personnel and that they only worked in that unit.

From the occupational therapy department, it was decided to include the resident in all the activities carried out in the psychogeriatric unit. Regarding different BPSD registered in relation to aggressiveness in grooming and dressing, aggressiveness towards other residents, and refusal to take medication, an intervention was carried out from the medical and psychological departments focused on the administration of crushed and camouflaged medication with meals, monitoring the correct intake by the medical and nursing department. Additionally, they intervened in facilitating access to her seasonal garments, removing non-corresponding garments from her wardrobe (it must be clarified that the clothes have a meaning in the patient’s life history, i.e., her work occupation, as a clerk in her own clothing store). To work on her spatio-temporal orientation, a system was implemented where the patient had visual access to different objects such as clocks, paintings, and TVs, which facilitated her orientation. The form of intervention was readjusted in order to work on the aspect of functional independence to the characteristics of the unit, since it has large areas for safe ambulation as well as rest areas. For this reason, she was included in the program of walks to promote her functional independence and the redirection of excess activity.

## 3. Results

From the applications of different tests in three different time periods (from January 2020 to August 2021), we found a worsening of cognitive impairment in the last assessment. There was also a worsening of the BADL, measured with the Barthel Index. The other parameters remained stable (Table 2).

Regarding the results of the observation carried out by the staff of the unit, it was found that the most relevant problems that the patient presented were sleep disturbance and nocturnal ambulation, and, to a lesser extent, reluctance to take medication, aggressiveness during grooming and dressing, and attempted assaults on other residents. We also observed that aggressive behaviors occurred, to a large extent, when taking medication. For this reason, starting in June 2021, crushed and camouflaged medication was prescribed with meals, with the appropriate monitoring carried out. In the following months, a reduction in BPSD was observed (Table 3).

Finally, decreased agitation and improved spatio-temporal orientation; levels of attention; and, to a lesser degree, apathy were observed.

## 4. Discussion

LTC centers have faced multiples challenges during the COVID-19 pandemic, including the higher risk and higher burden of the residents with BPSD [1,10,11,12]. Persons with BPSD present higher risks of severe COVID-19 infection due to the relation between frailty and dementia, a higher risk of severe neuropsychiatric symptoms related to delirium and encephalopathy, higher difficulty participating in screening tests, lower ability to report symptoms of infections, and higher difficulty adhering to infection control measures due to their difficulties in comprehension and remembering, and they may place themselves at higher risk of infection because of their lower abilities to maintain social isolation, to stay in one place, or to wear face masks [11]. Due to the deep impact of COVID-19 on the healthcare system, BPSD were not addressed as needed [12]. Cancellation of therapeutic activities, routine disruption, and suspension of visits and social activities have been especially challenging for these residents, forcing them to live in a stressful environment that does not fulfill their psychosocial needs [11,12]. Complementary methods to prevent BPSD consequences in the context of the COVID-19 pandemic have included direct feedback and coaching, modeling, discussions in small groups, and informative sheets providing basic information about the residents for the staff members [12]. The technology was also integrated in the interventions, although low digital competences can be an obstacle for this kind of resource [12,13]. In their study, Danilovich et al. found that a pre-existing culture of teamwork and flexibility to adopt new approaches facilitated the response to COVID-19 in LTC facilities, whereas low levels of digital literacy and low job satisfaction due to lack of face-to-face interactions hampered adaptation to the pandemic. 

In the case reported, when the resident was admitted to the LTC center, she presented BPSD, as well as severe cognitive impairment measured with the MEC (score 14), which determines advanced dementia, and a moderate to severe cognitive defect, measured with the GDS (score 5). She also presented memory impairment, difficulty in retaining recent information, attention deficit, severe temporal disorientation, and very basic spatial awareness, with impaired judgment and problem solving, and moderate dependence for BADL. The resident did not manifest awareness of deterioration and/or illness.

Taking into account the results of research on the repercussions of social isolation derived from the COVID-19 pandemic, in terms of its impact on mental health both in the general population [10] and in the older adults admitted to LTC centers [2], strategies were designed aimed at slowing down, as much as possible, the advancement of dementia in the lockdown situation, especially with regard to BPSD [9]. In addition, considering that functionality worsens in dementia if adequate stimulation is not performed, the medical and psycho-social staff of the LTC center considered it ideal for this resident to follow a personalized attention program [6].

It should be noted that despite the strict measures experienced in Spanish LTC centers during in the first wave of the COVID-19 pandemic, no worsening was observed in the case study resident, even in the 4 months following the end of the lockdown. The explanation for these results is based on individualized attention and stimulation; the patient’s low awareness of reality; social support during the confinement phase (video call program); and freedom in mobility through the assigned area, which allowed her brief interactions with other residents, as well as other activities already exposed, which acted as protectors of mental health and functionality. We could affirm that ICTs, through the video call program, were a very relevant social resource for the resident in that phase [2].

In the last months of 2020 and early 2021, the patient suffered a worsening of BPSD (aggressiveness at bath time and personal hygiene, attempts to attack other residents, etc.), symptoms that could be attributable, on the one hand, to the worsening of dementia, and, on the other, to the limitation of space to wander. We can say that because she always wandered through the same spaces, after a while, those distances were not enough, and therefore the resident tried to occupy private spaces or her access to certain spaces was denied/limited by preventive anti-COVID-19 measures. After an assessment of this situation by the medical and psycho-social team, in March 2021, it was decided to relocate the patient to the psychogeriatric unit, where more specialized care was offered. However, the results of the last follow-up carried out on the resident (July 2021) indicate a worsening in the cognitive and functional scores and in the basic activities of daily life.

## 5. Conclusions

The numerous challenges of the management of BPSD in the context of the COVID-19 pandemic have stressed the relevance of multidisciplinary, teamwork-based, and comprehensive approaches [12,13]. Measures such as the sectorization of the center, the limitation of spaces, the performance of relatively invasive tests such as taking PCR samples or antigens, the continued use of individual protection equipment, the absence of social contact, and the alteration of daily routines caused by COVID-19 predictably favored the deterioration of residents, including an increase in BPSD. In the case study described here, during the stay of the patient in the psychogeriatric unit, a series of strategies were implemented in order to minimize the effects of social isolation. In terms of promoting the resident’s social contact with reference persons, ICTs were the most relevant social resource. The measures implemented helped reduce, but did not eliminate, cognitive and behavioral deterioration in the patient.

## Figures and Tables

**Figure 1 geriatrics-07-00001-f001:**
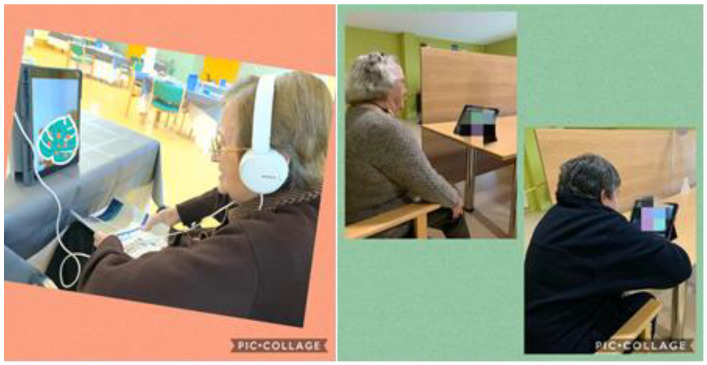
Video call program.

**Table 1 geriatrics-07-00001-t001:** Results of the MEC, GDS, Barthel, Yesavage, and CDR tests, applied on 31 January 2020.

Test
MEC:14
GDS: 5
Barthel Index: 85
Yesavage: 0
CDR: 2

**Table 2 geriatrics-07-00001-t002:** Follow-up in the results of the MEC, GDS, Barthel, Yesavage, and CDR tests.

31 January 2020	30 April 2020	30 July 2021
Test	Test	Test
MEC: 14	MEC: 14	MEC: 10
GDS: 5	GDS: 5	GDS: 6
Barthel: 85	Barthel: 85	Barthel: 80
Yesavage: 0	Yesavage: 0	Yesavage: 0
CDR: 2	CDR: 2	CDR: 2

**Table 3 geriatrics-07-00001-t003:** Incidents from July 2020 to August 2021.

	Sleep Disturbance and Nocturnal Ambulation	Poor Intake and Refusal to Take Medication	Aggression at Bath Time and Personal Hygiene	Attempts to Attack Other Residents
From July to October 2020	21 incidents	1 incident	0 incident	1 incident
From November 2020 to February 2021	27 incidents	5 incident	6 incident	2 incident
From March to May 2021	19 incidents	6 incident	9 incident	0 incident
From June to August 2021	2 incidents	0 incident	0 incident	0 incident

## Data Availability

All data will be made available upon reasonable request.

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
