# Peer review of "Behavioral Interventions in Long-Term Care Facilities during the COVID-19 Pandemic: A Case Study"

_geriatrics, 2021, doi:10.3390/geriatrics7010001_

Round 1

Reviewer 1 Report

The manuscript presents the description and evolution of a patient with cognitive impairment monitored first at home and then admitted to a geriatric unit, detailing the results of the assessment scales, the patient progression, the measures applied, etc.
Although the case description is complete and extensive, only one case is presented. For this reason, the scientific interest is less. The measures explained are part of the approach of these people during the Covid-19 pandemic. Could the authors present a series of cases related to the topic?
If the manuscript is finally published, Figure 2 should be removed, it has no scientific interest.

Author Response

Dear reviewer, 

Attached revision 1

Reviewer 2 Report

In the Case Report described,  Dosil-Díaz and colleagues described a case of a patient with dementia and BPSD, admitted in a long term care center, and described the how the restrictions due to COVID affected her cognittion and behaviour, as well as the management of her therapy, both pharmachological and cognitive. 

The case described could be interesting, however I recommend an extended revision of the English, in order to improve the quality of the reading.

Few examples: 

Line 42: SPCD: the acronym should be expanded, unless the Author refers to BPSD, in that case it should be corrected

Line 67: she without higher education... a verb is missing

Line 85: MED and GDS should be expanded in this line, where they have been first mentioned, please move from below in the Manuscript

Line 101: please repharse, otherwise "and one..." would seem another antipsychotic

Lines 110: the resident is described as "it" and afterwards "he", while the patient is a woman

Line 173: incomplete sentence

At the end of the Introduction, a sentence mentioning that the manuscript is a case report and a brief description would be necessary

Some literature about other LCT and other exeriences in patients with dementia should be reporteed in the discusison

Author Response

Dear reviewer,

Attached revision 2

Best regards

Round 2

Reviewer 1 Report

The authors have revised the manuscript, especially its edition in English, which has improved the presentation. The figure that was suggested has been withdrawn, and the impossibility of adding more cases is discussed.
If the manuscript is finally published, I suggest removing the acronym from the title of the manuscript "LTC" that did not appear in the first version, the title of a manuscript should not contain acronyms.

Other comments: Review the sentence: She did not attend a higher education institution (Material and methods section). 

The conclusions should be reviewed again, the interventions carried out are described again, and not their effects.

Author Response

Dear reviewer,

We want to thank the reviewer for his/her inputs. Following reviewer’s suggestions:

  • The acronym has been removed the title.
  • The sentence “She did not attend a higher education institution” has been removed and substituted for a more direct expression about the educational level of the person.
  • The conclusion has been shortened as much as possible, and a sentence on the effects of the intervention has been added.

best regards

Reviewer 2 Report

The Authors have improved the Manuscript. 

Only minor editing left (Line 81: a MEC, not an MEC).

Line 121-122 "a Spanish version of the Mini-Mental State Examination 101 with a total score of 35)" and  Global deterioration scale are not necessary in extenso anymore, since they have been moved in the previous section

line 173: the patient suffereds FROM a worsening in BPSD, or the patient EXPERIENCED a worsening in

Line 240: cancellation

Line 90: Memantin is not an antipsychotic

Author Response

Dear reviewer, 

Thanks for the comments and appreciations. Attached are the responses to your comments made regarding the manuscript.

Best regards,

Carlos Dosil Díaz
